

# Oxidative stress and the presence of bacteria increase gene expression of the antimicrobial peptide *aclasin*, a fungal CSαβ defensin in *Aspergillus clavatus*

Gabriela Contreras, Nessa Wang, Holger Schäfer and Michael Wink

Institute of Pharmacy and Molecular Biotechnology, Heidelberg University, Heidelberg, Baden-Württemberg, Germany

## ABSTRACT

**Background:** Antimicrobial peptides (AMPs) represent a broad class of naturally occurring antimicrobial compounds. Plants, invertebrates and fungi produce various AMPs as, for example, defensins. Most of these defensins are characterised by the presence of a cysteine-stabilised α-helical and β-sheet (CSαβ) motif. The changes in gene expression of a fungal CSαβ defensin by stress conditions were investigated in *Aspergillus clavatus*. *A. clavatus* produces the CSαβ defensin Aclasin, which is encoded by the *aclasin* gene.
**Methods:** *Aclasin* expression was evaluated in submerged mycelium cultures under heat shock, osmotic stress, oxidative stress and the presence of bacteria by quantitative real-time PCR.
**Results:** *Aclasin* expression increased two fold under oxidative stress conditions and in the presence of viable and heat-killed *Bacillus megaterium*. Under heat shock and osmotic stress, *aclasin* expression decreased.
**Discussion:** The results suggest that oxidative stress and the presence of bacteria might regulate fungal defensin expression. Moreover, fungi might recognise microorganisms as plants and animals do.

## INTRODUCTION

Antimicrobial peptides (AMPs) are a diverse group of naturally occurring molecules that are produced by a wide range of organisms, both prokaryotes and eukaryotes. AMPs are short peptides (consisting of 12–50 amino acids), commonly cationic and amphipathic. AMPs show a broad-spectrum activity against bacteria and other organisms, including gram-positive and gram-negative bacteria, parasites, fungi, viruses and even cancer cells (*Mahlapuu et al., 2016*). Additionally, AMPs are part of the innate defence mechanism in animals and plants (*Heine, 2008*). The role of AMPs is less understood in prokaryotes and lower eukaryotes. It has been suggested that AMPs might help them to compete for nutrients with other microorganisms (*Ageitos et al., 2017*).

Corresponding author
Michael Wink,
wink@uni-heidelberg.de

The largest group of AMPs are defensins. Defensins are cysteine-rich peptides found in many organisms throughout the eukaryotic kingdom. Vertebrate defensins contain a three-stranded antiparallel β-structure. Defensins produced by invertebrates, plants and fungi mostly contain a common structure composed of an α-helix linked to a β-sheet by three to four disulphide bridges. This structure is called cysteine-stabilised α-helical and β-sheet (CSαβ) motif (*Silva, Gonçalves & Santos, 2014*).

Defensin genes can be differentially or constitutively expressed in plants, vertebrates and invertebrates. In these three groups of organisms, defensins are produced as host defence response to bacteria and fungi (*Harder et al., 1997*; *Irving et al., 2001*; *Penninckx et al., 1996*). Signalling pathways of host defence response share similarities among plants, invertebrates and vertebrates (*Han et al., 1998*; *Ip et al., 1993*; *Ryals et al., 1997*). For example, the receptors, which recognise pathogens, are highly conserved among these organisms (*Medzhitov, Preston-Hurlburt & Janeway, 1997*; *Whitham et al., 1994*).

In plants, presence of pathogens and abiotic stress (salt, drought and cold) activate the plant immune system signalling cascade whose defence response includes the expression of defensins as well as other AMPs (*Campos et al., 2018*; *Taji et al., 2004*). It has been suggested that defensins are involved in stress adaptation in addition to the antimicrobial activity (*Do et al., 2004*; *Koike et al., 2002*).

Fungi are attractive sources of antimicrobial compounds. However, only few fungal CSαβ defensins have been characterised in terms of their antimicrobial activity. Some of these defensins are plectasin, copsin and eurosin, which are synthesised by *Pseudoplectania nigrella*, *Coprinopsis cinerea* and *Eurotium amstelodami,* respectively (*Essig et al., 2014*; *Mygind et al., 2005*; *Oeemig et al., 2012*). These fungal defensins have antimicrobial activity mainly against gram-positive bacteria.

Another class of defensins found in fungi is defensin with antifungal activity, also called defensin-like antifungal. The structure of defensin-like antifungals consists of five antiparallel β-sheets forming a β-barrel. The gene expression of defensin-like antifungals from *Aspergillus giganteus* and *A. niger* is altered by environmental stress conditions (*Meyer & Stahl, 2002*; *Meyer, Wedde & Stahl, 2002*; *Paege et al., 2016*). For example, carbon starvation increases the expression of these defensin-like antifungals (*Meyer, Wedde & Stahl, 2002*; *Paege et al., 2016*). In *A. giganteus,* osmotic stress, heat shock and the presence of other fungi enhance the gene expression of antifungals (*Meyer & Stahl, 2003*; *Meyer, Wedde & Stahl, 2002*). In *Penicillium chrysogenum*, limited glucose induces the expression of antifungals (*Marx et al., 1995*). In both cases, defensin-like antifungal genes contain putative stress response element (STRE) sequences in their promoters (*Marx, 2004*; *Meyer, Wedde & Stahl, 2002*). STRE is a consensus sequence (5′-AGGGG-3′ and 5′-CCCCT-3′) and is present in the promoters of genes that are regulated by stress.

In silico analysis indicated the identification of eight families of putative CSαβ defensins from published fungal genomes (*Zhu, 2008*; *Zhu et al., 2012*). This prediction was based on the sequence similarity and the presence of the CSαβ motif. Among the putative defensins described is Acasin (encoded by *acasin* gene), which is produced by the filamentous fungus *A. clavatus* (*Zhu, 2008*).

As mentioned previously, environmental conditions can alter the expression of plant defensins and some defensin-like antifungals from fungi. However, the transcriptional regulation of CSαβ defensins is not well-understood. Stress conditions might also change the gene expression of fungal CSαβ defensins, such as *aclasin*. *Aclasin* expression might be a defence response to bacterial presence. The present work aimed to investigate *aclasin* expression under stress conditions (heat shock, osmotic, oxidative and presence of bacteria) in submerged mycelium of *A. clavatus*. The present investigation demonstrates that oxidative stress and the presence of bacteria increased two-fold *aclasin* expression after 30 min of exposure. Our findings suggest that filamentous fungi would produce CSαβ defensins as defence response. Moreover *A. clavatus*, as other fungi, might possess specific mechanisms of bacterial recognition.

## MATERIALS AND METHODS

### Strains and conditions of culture

*Aspergillus clavatus* DSM 3410 and *Bacillus megaterium* DSM 32 were obtained from the German Collection of Microorganisms and Cell Cultures (DSMZ). *A. clavatus* was grown on malt extract broth (MEB; 2% malt extract, 2% glucose, 0.1% peptone) at 28 °C. *B. megaterium* was cultured on Luria-Bertani (LB) medium (0.5% yeast extract, 1% tryptone, 1% NaCl) at 37 °C.

For conidia production, *A. clavatus* was grown on MEB at 28 °C for 5 days. Conidia were harvested by washing MEB agar plates with five ml of phosphate buffered saline containing 0.01% Tween 80 and filtered through VWR 413 filter paper (VWR International BVBA, Leuven, Belgium). For submerged cultures of *A. clavatus*, a conidia suspension was inoculated in MEB to reach a final concentration of $10^5$ conidia/ml.

### Biomass and glucose determination

The biomass was determined from the vegetative mycelium of submerged *A. clavatus* after 12, 24, 36, 48, 72, 96 and 120 h of cultivation. Mycelia were filtered through VWR 413 filter paper (VWR International BVBA, Leuven, Belgium). The filter papers were washed with water and dried at 80 °C for 24 h.

Glucose in the medium was measured after 12, 36, 48, 72, 96 and 120 h of cultivation. Glucose was quantified based on the combined action of glucose oxidase (GOD) and peroxidase (POD) (*Barham & Trinder, 1972*). Briefly, two µl of each culture was incubated with 200 µl of GOD-POD reagent, which consists of 100 mM potassium phosphate buffer, pH 7.40, 10 mM phenol, 0.3 mM 4-aminoantipyrine (Fluka Chemie, Buchs, Switzerland), 10 kU/l GOD (Sigma-Aldrich, Saint Louis, MO, USA) and 700 U/l peroxidase (Sigma-Aldrich, Saint Louis, MO, USA). Samples were incubated for 30 min at 37 °C. Then, the absorbance was read at 505 nm using a TECAN spectrophotometer (Infinite 200 PRO NanoQuant, Grödig, Austria).

### Heat shock, osmotic and oxidative stress conditions

For heat shock conditions, submerged cultures of *A. clavatus* were incubated in MEB for 17 h at 28 °C and 150 rpm. Then, cultures were transferred to 37 or 47 °C. Aliquots

were taken after 30 and 60 min of incubation for subsequent RNA expression analysis. For osmotic stress conditions, submerged cultures were treated with one M NaCl final concentration. Samples were taken after 30 min of incubation. For oxidative stress, submerged cultures were treated with 2, 5 or 10 mM hydrogen peroxide ($H_2O_2$) based on previous studies in *Aspergillus* ssp. (*Fountain et al., 2015*; *Reverberi et al., 2008*). Samples were taken after 30 and 60 min of incubation with $H_2O_2$.

## Co-incubation with *B. megaterium*

*Bacillus megaterium* was grown in LB medium at 37 °C to an optical density at 600 nm of two exponential phase. The bacterial suspension was concentrated and stored at 4 °C for 30 min. For heat-killed culture, it was treated at 60 °C for 30 min. Colony-forming units (CFUs) were counted for each bacterial suspension after 16 h of incubation on LB agar plates at 37 °C.

*Aspergillus clavatus* was grown 17 h at 28 °C, 150 rpm. Viable ($10^7$ CFU/ml final concentration) and heat-killed bacteria were inoculated in submerged culture of *A. clavatus*. Samples were taken at 15 min after inoculation for later RNA expression analysis.

## RNA purification, cDNA synthesis and qPCR

Total RNA of three biological replicates was extracted. Mycelia were harvested by vacuum filtration using a VWR 413 filter paper (20 μm pore size; VWR International BVBA, Leuven, Belgium). Then, mycelia were washed with water and immediately frozen in liquid nitrogen. Samples were kept at −80 °C until RNA purification. Mycelia were disrupted by grinding with a mortar and pestle using liquid nitrogen. RNA was purified by FastGene RNA Premium Purification Kit (Nippon Genetics, Tokyo, Japan) according to the manufacturer's instructions. RNA was treated with DNase I (Nippon Genetics, Tokyo, Japan). The integrity of RNA checked by agarose gel electrophoresis. RNA was quantified spectrophotometrically at 260 nm. The synthesis of cDNA was performed using FastGene Scriptase II cDNA Kit (Nippon Genetics, Tokyo, Japan), two μg RNA and oligo (dT) according to the manufacturer's protocol.

Quantitative real-time PCR (qPCR) was carried out in triplicate in a LightCycler 96 instrument (Roche, Mannheim, Germany) using 100 ng cDNA, 0.5 μM each primer (Table S1) and qPCR BIO SyGreen Mix Lo-ROX (PCR Biosystem, London, UK) in 10 μl of total volume. The *act1* gene (ACLA_095800) was used as reference gene (*Bohle et al., 2007*; *Verheecke et al., 2015*). Threshold cycle ($C_t$) determination was performed using LightCycler 96 software (Roche, Mannheim, Germany). Expression levels were calculated according to the $2^{-\Delta\Delta Ct}$ method (*Schmittgen & Livak, 2008*). Results are presented as the mean of three independent data. Statistical analyses were performed in GraphPad Prism 5.0 (GraphPad Software, San Diego, CA, USA).

## In silico analysis for transcription factor binding sites

The 5′ untranslated region (UTR) of eight genes of fungal CSαβ defensins were analysed to identify putative transcription factor binding sites (TFBS) (Table S2). 1,500 bp upstream of the translation start site were analysed by the *Saccharomyces cerevisiae* promoter

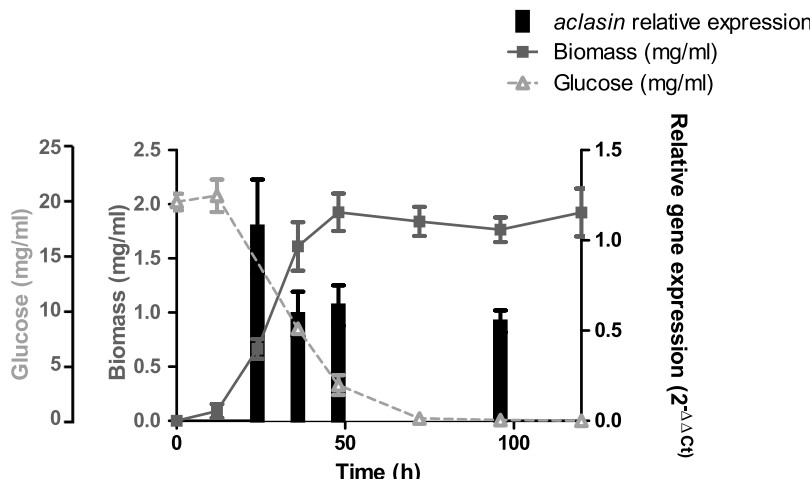

**Figure 1 Relative *aclasin* mRNA (ACLA_006820) expression during mycelial growth in *A. clavatus* DSM 3410.** *A. clavatus* ($10^5$ conidia/ml) was incubated in MEB at 28 °C and 150 rpm. Biomass represents the mycelial dry weight. For RNA analysis, mycelia samples were collected at 24, 36, 48, and 96 h of incubation. Relative *aclasin* mRNA expression was determined by qPCR according to the $2^{-\Delta\Delta Ct}$ method. The *act1* gene (ACLA_095800) was used as housekeeping gene. Normalised gene expression values for each time were compared with normalised gene expression at 24 h of incubation. Data are means and standard deviations from three independent experiments. *Aclasin* relative expressions were analysed by ANOVA followed by Bonferroni post-tests ($P > 0.05$).

database of the Cold Springs Harbor Laboratory (http://rulai.cshl.edu/SCPD/) (*Zhu & Zhang, 1999*) and the TRANSFAC 6.0 database using the Patch 1.0 program (http://gene-regulation.com/cgi-bin/pub/programs/patch/bin/patch.cgi) (*Matys et al., 2006*). The *act1* gene (ACLA_095800) of *A. clavatus* was included as negative control (GenBank accession number: NW_001517095.1; Region: 176194-177694).

# RESULTS

## *Aclasin* expression during vegetative mycelium growth

The *aclasin* gene (ACLA_006820) encodes Aclasin, a putative CSαβ defensin produced by *A. clavatus* (*Zhu, 2008*). *Aclasin* expression was detected by qPCR in different stages of the vegetative mycelium growth in submerged cultures: during exponential (24 h), late exponential (36 h), early stationary (48 h) and stationary growth phase (96 h). Simultaneously, the glucose content of the culture was determined during mycelium growth. The results of the *aclasin* gene expression are shown in Fig. 1. *Aclasin* was expressed in vegetative mycelium in these four stages, and no significant differences among them were observed. *Aclasin* expression did not change when glucose in the medium was depleted (96 h). These results indicate that steady-state level of *aclasin* does not change under glucose depletion.

## *Aclasin* expression under stress conditions

Submerged cultures of *A. clavatus* were subjected to heat shock, osmotic and oxidative stress. The *aclasin* gene expression was assessed by qPCR through the change in the mRNA level compared to the untreated culture.

### Heat shock and osmotic stress decreased aclasin expression

For heat shock conditions, *A. clavatus* was grown in liquid MEB medium for 17 h and transferred to 37 or 47 °C. Changes in the *aclasin* gene expression were evaluated after 30 and 60 min; the results are presented in Fig. 2A. The *aclasin* gene expression showed a significant decrease compared to the untreated control after 60 min: 10 and 2.8 fold at 37 and 47 °C, respectively. In addition, *hsp30* (ACLA_088240) expression analysis was included as a control because *hsp30* is upregulated by heat shock (*Seymour & Piper, 1999*). *Hsp30* encodes to a heat shock-like protein (HSP-30) that is responsible for the heat shock response in fungi (*Tiwari, Thakur & Shankar, 2015*). *Hsp30* expression was enhanced both at 37 and 47 °C, corroborating a heat shock response at transcriptional level. The lowest expression of *aclasin* was observed at 37 °C and 60 min of incubation. On the contrary, the highest expression of *hsp30* was observed in the same conditions.

Aclasin relative expression decreased eight and three fold at 37 °C and 47 °C, respectively. An increase of nine fold in *hsp30* relative expression was observed at 37 °C which suggests a stronger heat shock response at 37 than 47 °C in *A. clavatus*.

To investigate the expression of *aclasin* under osmotic stress conditions, submerged cultures of *A. clavatus* were treated with one M NaCl. *Aclasin* expression decreased four fold in comparison to the untreated control after 1 h of incubation, as shown in Fig. 2B.

### Oxidative stress increased aclasin expression

Oxidative stress conditions were generated by $H_2O_2$. The *aclasin* gene expression was evaluated at 2 mM $H_2O_2$ after 30 and 60 min (Fig. S1). In response to oxidising conditions, fungi produce different scavengers against $H_2O_2$, as the mycelial bifunctional catalase-peroxidase (*Paris et al., 2003*) that is encoded by the *cat2* gene (ACLA_044200). The *cat2* expression was assessed as a control to confirm the response to oxidative stress at transcriptional level (*Paris et al., 2003*). A significant increase in *aclasin* expression was observed after 30 min of incubation. No difference in *aclasin* expression was observed between cultures treated with 2 mM $H_2O_2$ and non-treated ones after 60 min.

Submerged cultures of *A. clavatus* were treated with higher concentrations of $H_2O_2$, 5 or 10 mM, and *aclasin* expression was evaluated (Fig. 2C). *Aclasin* expression increased two fold after 30 min of incubation ($P < 0.05$). Higher expression of the *cat2* expression was observed both at 5 and 10 mM $H_2O_2$, as expected.

### Aclasin expression under co-incubation with B. megaterium

To determine whether the presence of microorganisms can alter *aclasin* expression, *A. clavatus* was co-incubated with the gram-positive bacterium, *B. megaterium*. *A. clavatus*, after 17 h of incubation, was inoculated with a viable ($10^7$ CFU/ml) or a heat-killed *B. megaterium* suspension. Bacterial suspension was heat-inactivated to prevent nutrient depletion and basification of the medium due to bacterial growth (*Farrell & Finkel, 2003*). The results are shown in Fig. 3. The *aclasin* gene expression significantly increased two fold ($P < 0.01$ and $P < 0.05$, respectively) both in the presence of viable and heat-killed *B. megaterium* cultures.

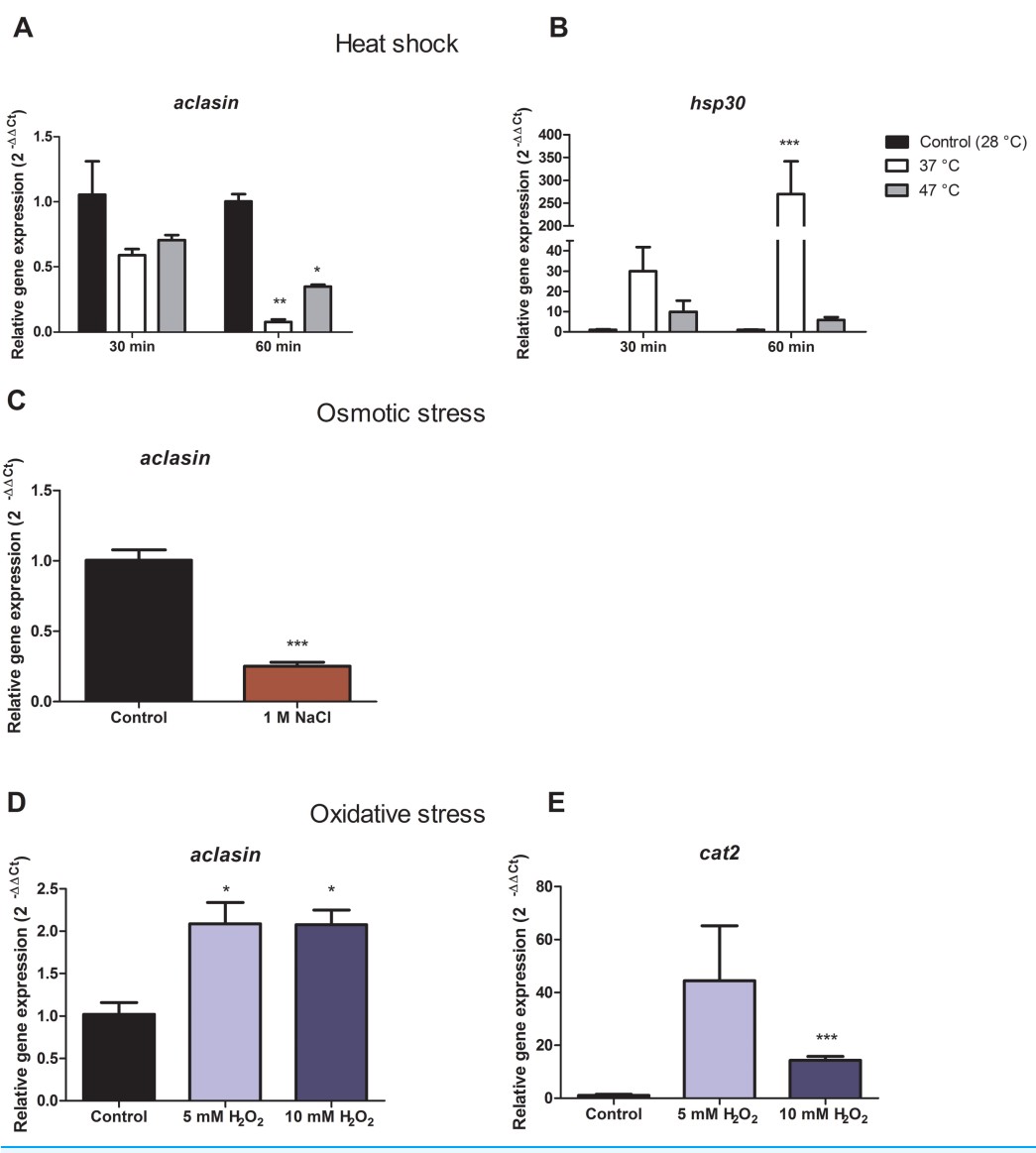

**Figure 2 Relative *aclasin* gene (ACLA_006820) expression under heat shock, osmotic, and oxidative stress conditions.** Submerged cultures of *A. clavatus* were incubated for 17 h in MEB at 28 °C and 150 rpm. (A) For heat shock stress, cultures were transferred to 37 or 47 °C. Aliquots were taken at 30 and 60 min. (B) *Hsp30* (ACLA_088240) was included as a positive control. (C) For osmotic stress, cultures were treated with 1 M NaCl for 1 h. (D) For oxidative stress, cultures were treated with 5 or 10 mM $H_2O_2$ for 30 min. (E) *Cat2* gene ( ACLA_044200 ), which encodes a mycelial bifunctional catalase-peroxidase, was used as a positive control for a gene upregulated by oxidative stress. Relative *aclasin* mRNA expression was determined by qPCR according to the $2^{-\Delta\Delta Ct}$ method. The *act1* gen (ACLA_095800) was used as reference gene and the untreated condition as control. Controls have a value of 1. Data represent the mean (± standard deviations) of three biological replicates. Relative expressions were analysed by ANOVA followed by Bonferroni post-tests in comparison to the control for heat shock and oxidative stress, and student t-test for the osmotic assay (*, $P < 0.05$; **, $P < 0.01$; ***, $P < 0.001$).

## Identification of putative regulatory elements in the 5′ untranslated region

The 5′ untranslated region of *aclasin* was analysed to identify putative regulatory elements. Several putative TFBSs were found, including TATA box, STRE, Yap1 response element

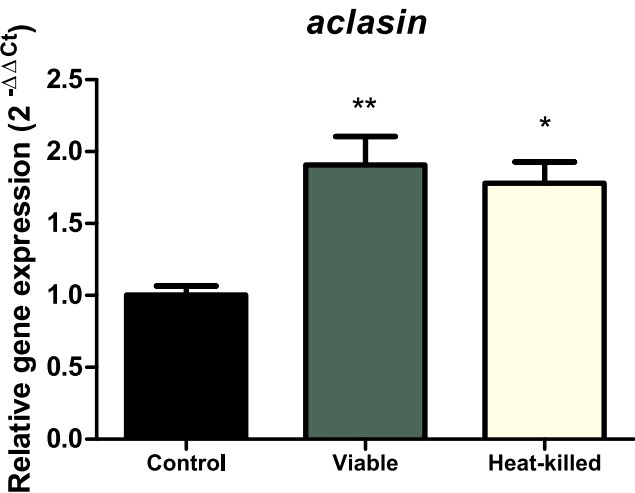

**Figure 3** Relative *aclasin* gene (ACLA_095800) expression in *A. clavatus* in co-cultivation with *B. megaterium*. *A. clavatus* was incubated 17 h in MEB at 28 °C and 150 rpm. Cultures were inoculated with viable ($10^7$ CFU/mL) or heat-killed culture of *B. megaterium*. Samples were taken after 15 min for total RNA purification. The *act1* was used as reference gene. Data represent the mean (± standard deviations) of three biological replicates. The relative expressions were analysed by ANOVA followed by Bonferroni post-tests (*, $P < 0.05$; **, $P < 0.01$). 

(YRE) and cAMP response element (CRE) (Fig. S2). In *Aspergillus* ssp., Yap1 is a transcriptional activator of associated genes to oxidative stress (*Hong, Roze & Linz, 2013*). CRE is recognised by transcriptional factors relating to oxidative and osmotic stress responses in *Aspergillus* sp. (*Balázs et al., 2010*; *Hong, Roze & Linz, 2013*). In addition, the 5′ UTR of the other seven CSαβ defensins (*Zhu, 2008*; *Zhu et al., 2012*) were analysed to identify STRE, CRE and YRE sequences. Putative STRE sequences were found in all of the 5′ UTR analysed (Fig. S3). Nevertheless, STRE is a short sequence, and it was also found in the 5′ UTR of the housekeeping *act1* gene that was included as a negative control.

## DISCUSSION

The transcriptional regulation of fungal CSαβ defensins was an open question. Understanding their transcriptional regulation could help us to comprehend the biological functions of AMPs and the interactions between microorganisms. Considering that the structure of CSαβ defensins is highly conserved, understanding their transcriptional regulation might help us to understand these functions in higher eukaryotes. Furthermore, *A. clavatus* (as other *Aspergillus sp.*) is a good model system for molecular biology studies because it grows on inexpensive, defined media and can be stored for long periods (*Golduran & Morris, 1995*).

In this study, *aclasin* expression increased about two fold after treatment with $H_2O_2$. Oxidative stress could directly or indirectly regulate *aclasin* expression. Previous studies have shown that a CSαβ defensin is involved in oxidative stress signalling (*Eigentler, Pócsi & Marx, 2012*).

All 5′ UTR of analysed defensin genes contained STREs sequences; however, it was also found in 5′ UTR of *act1*. *Act1* is considered a housekeeping gene, and its gene expression is stable under stressful conditions in fungi (*Jacob et al., 2012*). STRE is a short

sequence (5 bp) and can be found randomly in 5′ UTR of genes. Presence of putative STREs has been linked to gene regulation by stress in fungi without experimental data. Nevertheless, the implication of regulatory elements requires experimental analysis.

A significantly lower expression of *aclasin* was found at high temperatures and elevated osmolarity. In plants, an indirect down-regulation of defensin by heat shock factors has been observed (*Kumar et al., 2009*). This phenomenon also might occur in fungi. A possible explanation for this response is that many bacteria are not capable of surviving under high temperature and high osmolarity conditions. Filamentous fungi would not require the defensins against bacteria in these conditions.

As mentioned earlier, defensin-like antifungals are induced by glucose limitation (*Marx, 2004*). Moreover, the gene expression of a defensin-like antifungal, AFP, decreased under oxidative stress (*Meyer, Wedde & Stahl, 2002*). In contrast, the *aclasin* expression did not change under glucose limitation (stationary phase) in this study. Furthermore, *aclasin* expression was enhanced in the presence of $H_2O_2$. Although defensin-like antifungals and CSαβ defensins are AMPs, their gene expression patterns are not comparable.

The gene expression of a fungal CSαβ defensin produced by *A. nidulans* (*anisin1*) was associated with conidiation (asexual development) (*Eigentler, Pócsi & Marx, 2012*). The expression of defensin-like antifungals from fungi also relates to asexual development (*Meyer & Jung, 2018*). The correlation between *aclasin* expression and conidiation was not the focus of this work.

Invertebrate and fungal Csαβ defensins share structures (CSαβ motifs), mechanisms of action (*Schmitt et al., 2010*; *Schneider et al., 2010*) and have been proposed to have a common origin (*Zhu, 2008*). Their induction might also be similar. In this work, the presence of both viable and heat-killed *B. megaterium* increased *aclasin* expression two fold. In invertebrates, induction of defensins is triggered by bacterial recognition. Specifically, plants and animals recognise microbe-associated molecular patterns (MAMPs) (*Nürnberger et al., 2004*). *Aspergillus* and other fungi might detect MAMPs through physical interaction (*Schroeckh et al., 2009*; *Svahn et al., 2014*). Nevertheless, this detection mechanism has not been found in fungi.

Future research could be addressed to study *aclasin* expression after addition of peptidoglycan, lipoteichoic acid or other MAMPS to identify what trigger *aclasin* expression. Moreover, it would be attractive to study the *aclasin* expression under presence of fungi and gram-negative bacteria. Because, both fungi and gram-negative bacteria upregulate the gene expression of defensins in the red flour beetle, although these defensins are not active against these microorganisms (*Tonk et al., 2015*).

Previous studies have reported that in *A. giganteus*, co-cultivation with fungi induced expression of a defensin-like antifungal (*Meyer & Stahl, 2003*). This is the first report in which the presence of bacteria can induce a fungal CSαβ defensin. This would suggest a prominent role as antibacterial agent. We have shown previously that recombinant AfusinC (a CSαβ defensin from *A. fumigatus*), as well as other CSαβ defensins, exhibit prominent antibacterial activity against gram-positive bacteria (*Contreras et al., 2018*; *Essig et al., 2014*; *Mygind et al., 2005*).

## CONCLUSIONS

In this work, we demonstrated that *aclasin* was expressed in submerged mycelia of *A. clavatus*. The gene expression did not significantly change between 24 and 96 h of mycelial growth, which suggests that it is constitutively expressed in mycelia. Moreover, oxidative stress and the presence of bacteria slightly enhanced the *aclasin* mRNA expression. It would be interesting to investigate the corresponding protein levels of Aclasin under these conditions. Our results suggest that oxidative stress and the presence of bacteria could, directly or indirectly, regulate *aclasin* expression. These stress conditions could also induce the gene expression of other CSαβ defensins.

## ACKNOWLEDGEMENTS

We thank Mariana Roxo for her contribution in experimental design, and Hedwig Sauer-Gürth and Malak Dimirieh for their suggestions about RNA isolation.

### Funding

Deutsche Forschungsgemeinschaft provided financial support within the funding programme Open Access Publishing, by the Baden-Württemberg Ministry of Science, Research and the Arts and by Ruprecht-Karls-Universität Heidelberg. Gabriela Contreras was funded by a CONICYT/BECAS CHILE scholarship (72150082). The funders had no role in study design, data collection and analysis, decision to publish, or preparation of the manuscript.

### Grant Disclosures

The following grant information was disclosed by the authors:
Baden-Württemberg Ministry of Science, Research and the Arts and by Ruprecht-Karls-Universität Heidelberg.
CONICYT/BECAS CHILE scholarship: 72150082.

### Competing Interests

Michael Wink is an Academic and Section Editor for PeerJ.

### Author Contributions

- Gabriela Contreras conceived and designed the experiments, performed the experiments, analyzed the data, prepared figures and/or tables, authored or reviewed drafts of the paper, approved the final draft.
- Nessa Wang conceived and designed the experiments, authored or reviewed drafts of the paper, approved the final draft.
- Holger Schäfer authored or reviewed drafts of the paper, approved the final draft.
- Michael Wink conceived and designed the experiments, contributed reagents/materials/analysis tools, authored or reviewed drafts of the paper, approved the final draft.

## Data Availability

The raw data are available as a Supplemental File. The raw data shows the qPCR Cycle Threshold (Ct) mean values of the housekeeping gene (*act1*) and the genes tested (*aclasin, cat2, hsp30*).

Ct mean was calculated using three technical replicates.

## Supplemental Information

Supplemental information for this article can be found online at http://dx.doi.org/10.7717/peerj.6290#supplemental-information.

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
