# Peer review of "Oxidative stress and the presence of bacteria increase gene expression of the antimicrobial peptide aclasin, a fungal CSαβ defensin in Aspergillus clavatus"

_PeerJ, doi:10.7717/peerj.6290_

## Round 0.1 · original submission · Minor Revisions

Both the reviewers and myself consider your manuscript of interest deserving publication. However, there are some amendments and technical questions that have been requested and have to be taken into account; also, some details in the introduction and experimental details should be clarified in method section to improve quality of the paper.

Reviewer 1 ·

Basic reporting

The authors in the submitted manuscript titled “Oxidative stress and the presence of bacteria increase gene expression of the antimicrobial peptide aclasin, a fungal CSαβ defensin in Aspergillus clavatus” presents information which I believe will be of interest to many researchers in the field of microbiology, especially those with interest in the area of anti-microbial peptides. I would like to commend the authors for their work. However, there are several shortcomings which needs to be addressed as outlined below:
1. While I appreciate the general background information provided in the introduction section, it lacks details in some sections. Most of the citations in this section are from review articles, book chapters etc. While reviews and book chapters are good source of general information, lack of direct inputs from original research papers makes this section of the manuscript very vague in some places. For example, in lines 46-47 the authors have merely mentioned how presence of pathogen or abiotic stress factors induce defensin gene expression without elaborating the significance of this information. Explaining the link between abiotic stress and need for expression of anti-microbial defensins would allow the readers to better understand the physiological significance of such anti-microbial peptide production. It is also true for lines 43-46. Additionally, in some places like lines 46-47 and 56-58, it is not clear whether the statements are true for all organisms within the group mentioned (plants, fungi etc.) or, are only true for some organisms within the group.
2. In the last paragraph of the introduction section, the authors have summarized their statements from the previous paragraphs of the section and have attempted to rationalize their endeavor for the experiments undertaken. However, a compelling statement to justify the motivation of the proposed study is lacking. The authors should consider modifying this part to summarize their motivation, findings and significance of their study in broader context more effectively and clearly.
3. In lines 43-46, it will be better to rephrase the sentence “…microorganisms can induce defensin gene expression through bacterial recognition through receptors…” to convey the message more clearly.
4. Line 18: The abbreviation “(CSαβ)” should be introduced at the end of the sentence “… cysteine-stabilised α-helical and β-sheet motif” before it is mentioned stand alone in the next sentence.
5. Line 41: “…structure composed by an α-helix…” should be “…structure composed of an α-helix…”
6. Line 48: “however, few fungal” could be better written as “however, only few fungal”.
7. Line 120: “checked through an agarose gel” could be better written as “checked by agarose gel electrophoresis”
8. Lines 134-135: it could be better written as “…upstream of the translation start site were…”
9. Line 165: I believe the authors meant “aclasin” instead of “afusin”.
10. Line 187: it could be better written as “…both in the presence of viable and…”
11. Some minor formatting issues with the references like incorrect or missing page numbers in lines 306, 338, 351 and 358.

Experimental design

1. The authors have studied the effect of oxidative stress on aclasin expression using 5 or 10 mM H2O2 (line 172). It is not clear why such concentrations were chosen? Is it known previously that such concentrations induce oxidative stress? It is particularly important because the authors also used a lower concentration of 2 mM later (in lines 178-179). The motivation for checking the lower concentration is not clear.
2. In addition to the live culture of B. megaterium, the authors have also used heat-killed bacteria. The inspiration of checking the effect of heat-killed bacteria on aclasin expression is not clear.

Validity of the findings

1. Lines 210-214: the authors have mentioned that they also found STRE sequences in the 5’ UTR of the act1 gene which was included in the study as a negative control. The authors should reconcile how the presence of this sequence in the 5’UTR of the act1 gene affects their observation in this regard. The authors have themselves written in lines 212-213 that act1 is a house keeping gene whose levels are stable under stressful conditions. If this is true, then should the presence of STRE sequences alone be interpreted as a signature of stress induced regulation of gene expression? Should we call act1 as a “negative control”? Does it invalidate the findings by the authors in this regard?
2. In Figure 1, if we look at the supplemental excel file provided by the authors, the glucose concentration at 24h time point is recorded as “0”. This is bit surprising that the glucose levels would be zero at that time point whereas it is present in the previous and succeeding time points. Is it an experimental error? Or, the glucose concentration was not determined at this time point?
3. In the lines 150-151, the authors claim that the glucose reduction does not influence aclasin expression. Based on the data from Figure 1, I think this conclusion is a bit inaccurate. The technique used in this study (qPCR) only measures the steady state levels of mRNA present at that point and not transcriptional control. Hence the conclusion should be modified to state that the “steady state levels of aclasin does not change with glucose depletion”.
4. In Figure 2A, aclasin expression at 47 C from 30 min to 60 min reduce by ~50%, whereas the same for 37 C reduces by ~3-4 fold. The authors need to discuss the significance and/or explanation of this observation in their results section.
5. Figure 2C: the authors do not provide any possible explanation for the observed reduction in the levels of cat2 from 5 mM to 10 mM H2O2. Ideally, we would expect stronger induction of cat2 levels in higher concentrations of the oxidant.
6. The authors need to reinforce the discussion section by mentioning how their study fits into the broader picture in the field. Does their finding lead to further questions which should be addressed in future?

Reviewer 2 ·

Basic reporting

• The manuscript is professionally written and the reading seems fluid for not expert in microbial molecular biology also. It conforms to journal standards. The language is unambiguous at all.
• The literature is sufficient referenced.
• Fig.1 _ What’s the experimental condition that was set as control and allowed to calculate and compare the levels of the relative expression of aclasin (2-ΔCt method) at the different time points? This is not clear. If the plot represents instead an absolute quantification among all the samples, please correct the graph axes and figure 1 title.
• Fig. 1 Legend _ Even though there are not significant differences, the performed statistic method is not indicated. It should be mentioned together with the p value threshold, as the authors already did for the rest of the figures.
• Figure 2 panel A _ aclasin graph, please indicate that the black bar of the Control (28 °C) used for relative quantification is =1 for the 60 minutes’ condition. I guess so because the black bar average at 30 minutes is higher than 1.0 in y-axis
• Line 165 _ ‘afusin’ level is not present in the figure. I guess it was written by mistake instead of ‘aclasin’. In fact, there are not following comments on afusin results or primers listed in the submitted table. Please correct it.

Experimental design

• 17 hrs for A. clavatus is the time of culture before treating with different kind of stress. For co-incubation experiment the authors used the bacteria after 16 hrs + 30 minutes. Are the bacteria in exponential phase growth at that time? Please comment it
• About heat-killing of B. megaterium, the authors should provide a brief explanation or a reference or previous experimental data that assure there were not bacterial spores at the time of killing at 60°C.
• It would be nice if the authors confirm in the main text that 2 mM-10mM concentrations are subletal (or including previous citotoxicity exclusion data as supplementary info). This to exclude a decrease in mRNA expression due to cell death.
• Regarding the co-incubation experiment, did the investigators separate bacteria from fungi before extracting RNA? Or bacterial RNA from the eukaryotic RNA? If yes, please write how in material and method (strainer? Filter paper cut off? other?), on the contrary it would be necessary to describe or to show supplementary data about the specificity of the primers used for the PCR starting from mixed RNA sample.

Validity of the findings

• The paper is technically sound. Results are expressed clearly. the study design is organically structured and the sections are well linked each other.
• As the authors highlighted the fungal defensin transcriptional regulation during bacterial interaction is poorly understood, thus the research question is interesting.
• Data are well presented. Statistics is convincing. Speculation and conclusion are properly confined within the results presented.

Additional comments

It is a paper with basic RNA-only focus, thus the results are not soon useful to the scientific community. On the other hand, the manuscript is well-written and the topic of the research is very attractive for future biotechnological applications.
The data are robust. The methods are good. The limits of the study are clearly described by the authors, but the results are promising and encouraging downstream projects.

---

## Round 0.2 · accepted · Accept

Thanks for taking into consideration all comments made by the reviewers. Your manuscript is now ready for publication.

# Reviewer 1 ·

Basic reporting

The revised manuscript is very well written with appropriate references, figures and logical conclusions.

Experimental design

The authors have successfully addressed all the concerns.

Validity of the findings

The authors have successfully addressed all the concerns.

Additional comments

I would like to convey my best regards to the authors for their effort and work.

Reviewer 2 ·

Basic reporting

The additional references now included in the manuscript makes the background and the whole paper more thorough than the previous version.
The figures have been improved and corrected following the reviewers' suggestions.

Experimental design

Methods section is now accurate and unambiguous.

Validity of the findings

The results and the discussion have been improved as well.

Additional comments

The authors were able to reply critically the reviewers and all their comments were addressed.